# Obstetric Results after Fertility-Sparing Management of Non-Epithelial Ovarian Cancer

**DOI:** 10.3390/cancers15164170

**Published:** 2023-08-18

**Authors:** Szymon Piątek, Iwona Szymusik, Piotr Sobiczewski, Wojciech Michalski, Magdalena Kowalska, Mariusz Ołtarzewski, Mariusz Bidziński

**Affiliations:** 1Department of Gynecologic Oncology, The Maria Sklodowska-Curie National Research Institute of Oncology, 02-781 Warsaw, Poland; sobiczewskipiotr7@gmail.com (P.S.); wojciech.michalski@pib-nio.pl (W.M.); magdapetryka@o2.pl (M.K.); bidzinski.m@gmail.com (M.B.); 2Faculty of Medical Sciences and Health Sciences, Kazimierz Pulaski University of Technology and Humanities in Radom, 26-610 Radom, Poland; 3Department of Obstetrics, Perinatology and Neonatology, Center of Postgraduate Medical Education, 80 Ceglowska Street, 00-001 Warsaw, Poland; iwona.szymusik@gmail.com; 4Institute of Mother and Child, 01-211 Warsaw, Poland; mariusz.oltarzewski@imid.med.pl

**Keywords:** non-epithelial ovarian cancer, germ cell tumor, sex cord-stromal tumor, fertility-sparing surgery, obstetric outcome, birth rate, recurrence

## Abstract

**Simple Summary:**

Fertility-sparing treatment (FST) is the gold standard for the majority of young women with non-epithelial ovarian cancer (NEOC). Its rarity and wide histological diversity lead to difficulties in assessing the oncological and reproductive outcomes. The aim of the study was to assess the recurrence rates and obstetric results of patients with NEOC. In a group of 146 patients, there was no difference in disease-free survival between the women with sex cord-stromal tumors (SCST) and germ cell tumors (GCT). The recurrence risk in the first two years after treatment exceeded the chance of childbearing. The cumulative incidence rate of childbearing rose continuously since the diagnosis. Chemotherapy was not related to the chance of having a child. FST can be offered to young women with NEOC regardless of their histology (SCST vs. GCT); however, pregnancy should be delayed until 2 years after receiving the treatment due to the increased risk of recurrence.

**Abstract:**

Purpose: To assess the recurrence and birth rates among patients with non-epithelial ovarian cancer. Methods: The study included 146 patients with germ cell (GCT, n = 84) and sex cord-stromal tumors (SCST, n = 62), who underwent fertility-sparing surgery. Adjuvant chemotherapy was administered to 86 (58.9%) patients. Most cases (133 out of 146) were staged FIGO I. Results: The 5- and 10-year disease-free survival rates were 91% and 83%, respectively. The recurrence risk was not associated with tumor histology, stage or age. Twenty-four months after the treatment, the rate of recurrence was higher than the rate of childbearing. The childbearing rates kept rising after the treatment and exceeded the rate of recurrence after 2 years. The cumulative incidence rates of birth 36, 60 and 120 months after treatment were 13.24%, 20.75%, and 42.37%, respectively. Chemotherapy was not related to childbearing. The patients’ age was related to the chance of childbearing. Conclusions: The prognoses of GCT and SCST are similar. Close follow-ups along with contraception should be offered to women during the first two years after treatment due to the increased risk of recurrence. After this period, relapses are rare and women can safely become pregnant.

## 1. Introduction

Malignant forms of germ cell ovarian tumors (GCT) and sex cord-stromal ovarian tumors (SCST) constitute a vast majority of non-epithelial ovarian cancers (NEOC); however, they account for approximately 5–10% of all ovarian malignancies [1,2,3]. The heterogeneity of NEOC is extraordinary, with a wide spectrum of clinical presentations, histologies, and biomarkers [4]. Furthermore, the etiology and molecular mechanism of NEOC are largely unknown [5]. Nonetheless, the management of various types of NEOCs is similar and includes upfront surgery and adjuvant chemotherapy. Platinum-based chemotherapy remains the standard first-line systemic treatment, with the BEP (bleomycin, etoposide, and cisplatin) regimen being most commonly used in clinical practice [1].

Fertility-sparing treatment/management (FSM) meets the expectations of young women, who desire to conceive in the future, as fertility preservation is recognized as an important aspect of their quality of life [6]. Nowadays, it is introduced into the routine management of cervical and endometrial cancers, and occasionally, it may also be offered to women with rare gynecologic tumors [7]. FSM became the current standard of care also for women with NEOC, as many cases are diagnosed in women younger than 30 years of age [6]. Although FSM preserves the female genital tract, it is the anti-cancer treatment that decreases women’s reproductive potential through the loss of ovarian tissue and the potential gonadotoxicity of chemotherapy.

This study was designed to assess survival and reproduction in young women with NEOC, who had undergone a fertility-sparing treatment.

## 2. Materials and Methods

Patients with NEOC were retrospectively enrolled in the study. They were managed at the Department of Gynecologic Oncology, Maria Sklodowska-Curie National Research Institute of Oncology (MSCNRIO), from January 2000 to December 2020. The eligibility criteria for the study were as follows: (1) having a histopathological diagnosis of NEOC, (2) an age ≤ 40, (3) and having undergone fertility-sparing surgery. The exclusion criteria included having undergone bilateral salpingo-oophorectomy and having genetic abnormalities. The study flowchart is presented in Figure 1.

All of the women were provided with all of the information regarding the potential risks of radical and conservative management.

The histopathological examination was performed by a local pathologist. In more difficult cases, slides were re-review by pathologists from MSCNRIO. The classification of the International Federation of Gynecology and Obstetrics (FIGO 2014) was used to determine the stage of the disease in all patients. The fertility-sparing surgery (FSS) was defined as sparing the uterus and at least a portion of an ovary (cystectomy, unilateral salpingo-oophorectomy). Pregnancies that resulted in a live birth were assessed. The birth rate was defined as the number of patients who gave birth to a live newborn (>22 weeks of gestation) divided by the total number of patients. Miscarriages were not taken into account.

The follow-up after the fertility-sparing treatment included: (1) gynecologic examination with transvaginal ultrasonography every 3–4 months during the first 2 years and afterwards every 6 months up to 5 years following the treatment; and (2) biomarkers serum level measurements (inhibin B and/or estradiol in SCST; AFP, hCG, LDH, and CA19.9 in GCT). Computed tomography or magnetic resonance were ordered in cases of suspected recurrence. After 5 years of follow-up, patients were referred to a general outpatient gynecologist.

Disease-free survival (DFS) was measured from the date of first surgery to the date of recurrence.

In Poland, all newborns are obligatorily screened for congenital metabolic disorders. Gender, gestational age, date of delivery, birth weight, and Apgar score data were gathered at the Screening Test Department of the Institute of Mother and Child in Warsaw. All Polish citizens have an individual PESEL ID (11-digit personal identification number). Numbers and dates of deliveries were extracted from an electronic database at the Institute of Mother and Child using the PESEL ID of every woman included in the study. Data regarding conception ending in miscarriages were not available.

### 2.1. Ethics Approval

All patients provided written informed consent for the chosen treatment. All procedures were conducted according to the Declaration of Helsinki for Medical Research involving Human Subjects. The bioethics committee of Kazimierz Pulaski University of Technology and Humanities approved the study (KB/04/2023). The clinical decisions concerning the treatment were not influenced by the purpose of this paper.

### 2.2. Statistics

Standard descriptive statistics tools were used; frequency tables were collected for the categorical variables, extreme values, mean values, and standard deviations were collected for the continuous variables with a normal distribution, and extreme values and quartiles were collected for the continuous variables with another distribution. Survival curves were calculated using the Kaplan–Meier method and compared using the logrank test. The competing risk methodology was used to estimate the cumulative probability of birth after the treatment and disease recurrence. The results are illustrated in the cumulative incidence curves. All estimates are given with a 95% confidence interval. All tests were performed at the statistical significance level of 0.05. The statistical analysis was carried out using the IBM SPSS Statistics 23 package.

## 3. Results

The inclusion criteria of 146 consecutive patients are shown in Figure 1. The median age at diagnosis was 28 (range 17–40 years of age). Nulliparous women constituted the majority of the patients (n = 123; 84.25%). The median follow-up was 63.34 months (95%CI: 54.23–72.46 months). Detailed characteristics of the patients are shown in Table 1.

Bilateral ovarian tumors were found in 12 (8.22%) patients; however, only 1 (0.68%) of them was diagnosed with non-epithelial ovarian cancer on both sides (FIGO IB). In 11 patients, NEOC was diagnosed on one side, while the contralateral lesions were benign. The surgical treatment of bilateral lesions included a one-sided adnexectomy with tumorectomy/cystectomy on the other side or bilateral tumorectomy/cystectomy. Unilateral adnexectomy was performed on the side with the larger tumor or more suspicious lesion during pre- and intraoperative evaluation. An appendectomy was performed on eight (5.48%) patients; however, the histologic examination did not reveal cancer infiltration of the appendix in any case.

### 3.1. Surgical Management

The primary surgery was conducted in our department on 17 (11.64%) patients, while 129 (88.36%) women underwent surgeries in other hospitals and were afterwards referred to the department with the histological diagnosis of NEOC. In 77 out of 129 (59.69%) cases, the histopathological examinations were re-evaluated by pathologists with expertise in ovarian malignancy. In 39 (50.65%) women, the diagnosis was fully confirmed, in 29 (37.66%) it was specified, while in 9 (11.69%) cases, the diagnosis was changed.

Three (2.05%) patients underwent urgent surgery due to a tumor rupture and intraperitoneal hemorrhage. An adnexal tumor was diagnosed during pregnancy in 10 (6.85%) women. Three (2.05%) patients underwent surgery in the second trimester, while in seven (4.8%) cases, the surgical treatment was performed at the time of the cesarean section.

Almost all the patients had complete (no residual disease) surgery, apart from one (0.68%) patient with granulosa cell tumor FIGO stage IIIC, who had an incomplete macroscopic resection.

Restaging surgery was performed on 26 (17.81%) women (Table 2). It included omental and peritoneal biopsy, and peritoneal washing in all cases; pelvic lymphadenectomy on 14 patients, while contralateral ovarian biopsies were conducted on eight patients, and paraaortic lymphadenectomy was conducted on one patient. Unilateral adnexectomy was performed on eight patients (six with SCST, two with GCT), who had ovarian cystectomy/tumorectomy during primary surgery. No serious complications were recorded after the restaging procedures. Three (11.54%) women were up-staged: one with granulosa cell tumor from IA to IC3, one with dysgerminoma from IA to IC3, and one with dysgerminoma from IC to IIB.

In total, 75 (51.37%) patients underwent peritoneal staging (peritoneal biopsy, peritoneal washing, omental biopsy) and 33 (22.6%) retroperitoneal lymph node assessment.

### 3.2. Adjuvant Treatment

Chemotherapy was administered to 86 (58.9%) patients. The most common regimen was BEP (bleomycin, etoposide, and cisplatin), which was given to 81 (94.18%) women. One (1.19%) patient had a serious allergic reaction to bleomycin during the first cycle; therefore, she received EP regimen. Other first-line regimens were used much less frequently: VIP (etoposide, ifosfamide, and cisplatin) was used on two (2.38%) patients, CBDCA-TXL (carboplatin and taxol) was used on one (1.19%) patient and PVB (cisplatin, vinblastine, and bleomycin) was used on one (1.19%) patient.

The majority of the patients with GCT (76.19%; 64 out of 84) received adjuvant chemotherapy. Patients with GCT who were not treated with chemotherapy were staged IA (21.43%; 18 out of 84) and IC (2.38%; 2 out of 84). Among the patients with SCST, adjuvant systemic chemotherapy was administered in 22 out of 62 (35.48%) patients. Detailed relationships between staging, histology, and adjuvant chemotherapy are presented in Table 3.

The majority of patients received three and four cycles of chemotherapy, representing 23 (26.74%) and 43 (50%) patients, respectively. Five and six courses of chemotherapy were administered to 3 (3.49%) and 16 (18.6%) patients, respectively.

One out of three women who underwent surgery during pregnancy received two cycles of CBDCA-TXL during gestation and four cycles after delivery. The other two patients did not undergo any adjuvant therapy.

One patient with granulosa cell tumor FIGO IC3 refused chemotherapy. During the multidisciplinary team meeting, she was instructed to undergo a strict follow-up every 3 months.

A complete response after the treatment (surgical or surgical and adjuvant) was achieved in 144 patients. Two patients progressed during BEP chemotherapy.

### 3.3. Survival Analysis

A recurrent disease was diagnosed in 17 (11.64%) patients. The 5- and 10-year DFS for all included women were 91% and 83.6%, respectively. The recurrence most often affected patients with mixed GCT (21.05%; 4 out of 19), followed by those with a granulosa cell tumor (13.04%, 6 out of 46), dysgerminoma (11.54%, 3 out of 26), yolk sac tumor (9.09%, 1 out of 11), immature teratoma (8%, 2 out of 25), and Sertoli–Leydig cell tumor (7.69%, 1 out of 13). However, there were no differences in DFS between patients with sex cord-stromal and germ cell tumors (*p* = 0.937; Figure 2A). Alongside histology, age (*p* = 0.674) and FIGO staging also did not significantly influence the recurrence rates (*p* = 0.228, Figure 2B).

No relapses occurred among 26 patients after the restaging surgery. However, there was no statistical significance in the DFS between patients who underwent the restaging surgery compared with those who did not undergo the restaging surgery (Figure 2C).

### 3.4. Obstetric Outcomes

In total, 66 patients gave birth to at least one child after treatment (birth rate 45.2%; Table 4). The total number of children born to patients with NEOC was 106, of whom 96 were born during the follow-up period and 10 were born during the peri-surgical period (Table 5). Almost all of them were singleton gestations, with the exception of one pair of twins delivered 11 years after the treatment. Preterm deliveries were significantly more frequent among women diagnosed during pregnancy than they were among patients who conceived after the treatment was completed (50% vs. 6%, *p* < 0.001).

Neither the staging, nor chemotherapy influenced the chance of childbearing (Figure 3A,B). To assess the relationship between the patients’ age at the moment of diagnosis and chance of childbearing, the patients were divided according to cut-off points of 0.33 and 0.66. Three age groups were created: <22.6, ≥ 22.6 to <29.8, and ≥ 29.8. It was found that patients younger than 22.6 at the moment of diagnosis were more likely to have a child after the treatment than the older patients (Figure 3C).

Among the patients with recurrence, only one with dysgerminoma delivered a child 33 months after recurrence.

### 3.5. Competing Risks Analysis

The competing risk analysis showed that women with NEOC were more likely to conceive than they were to experience cancer recurrence within 10 years of the initial surgery (Figure 4). The 3-, 5-, and 10-year cumulative incidence rates of subsequent childbearing vs. recurrence were 13.24% (95%CI: 7.44–19.4) vs. 8.71 (95%CI: 3.97–13.45), 20.75% (95%CI: 13.42–28.09) vs. 10.55% (95%CI: 5.25–15.84), and 42.37% (95%CI: 31.39–53.34) vs. 13.76% (95%CI:6.97–20.55).

## 4. Discussion

The evaluation of the oncological and obstetric results of NEOC patients is still insufficient, as it mostly relies on retrospective studies with small groups of patients. Surgery is the first therapeutic approach to treating NEOC. According to the ESGO/ESMO guidelines, FSS should become a standard of care in the early stages of NEOC among young women [1]. Most NEOCs are diagnosed at an early stage and therefore have an excellent prognosis and 5-year survival rates exceeding 90% [8]. In our study, more than 90% of women were <36 years old and mostly (84.25%) nulliparous; therefore, fertility preservation was of note. Although such an approach is not recommended at more advanced stages of SCST [6], it may be appropriate in selected cases of GCT with low volume disease and a strong desire to preserve fertility. Our data are concordant with the literature: 133 out of 146 women who qualified for FSS were diagnosed with FIGO stage I tumor, while the disease-free survival after 5 years was estimated at 91% regardless of the histological type. Among the patients with more advanced cases (FIGO stage ≥ II), but a low volume disease, and those who underwent macroscopically radical surgery, we did not find difference in the DFS. However, the number of those patients was very small (SCST n = 2, GCT n = 11). The analysis of 28 patients with GCT in advanced stage FIGO IIIC, who underwent FSM, demonstrated good oncological outcomes, with a 12% recurrence rate and a DFS of 88% [9]. In the advanced stages of immature teratoma FIGO II/III treated with FSS, the reported 5-year DFS and overall survival rates were 69% and 89.9%, respectively [10].

Unlike epithelial ovarian cancer, most cases of NEOC are diagnosed at an early stage. In many cases of NEOC, the primary surgery is performed in the general gynecologic department and involves the resection of the ovarian cyst/tumor or unilateral adnexectomy. The histopathological diagnosis of NEOC is often unexpected and patients are referred to cancer centers for further management (observation vs. restaging surgery vs. chemotherapy). These patients often undergo improper preoperative management. Therefore, imaging and serum biomarker assessments should be performed. Due to rare incidence and difficulties in pathologic examination, it is necessary to confirm the diagnosis by an expert in ovarian pathology. Such management is time-consuming; in our study, the mean time between the primary and restaging surgeries was 63 days. Considering the long period between the primary surgery and all the examinations, the decision on restaging surgery should take into account the possible benefits for the patient. The patient’s preferences are also important in decision making.

The restaging surgery involves two issues: peritoneal procedures and lymph node assessments. Peritoneal staging is important for SCST, because these tumors mainly spread intraperitoneally [11]. The European Society of Medical Oncology indicates that performing a lymphadenectomy on SCST patients is not recommended because the incidence of lymph node metastases is low [12,13], whereas, in GCT, the significance of lymphadenectomy is not clear. Lymph node metastases in GCT are found in 18.1–29% of patients [14,15], but several studies have not demonstrated a beneficial effect of lymphadenectomy on patients’ survival [16,17]. This may be due to the high chemosensitivity of GCT and the fact that patients not undergoing initial nodal staging surgery can be safely cured using chemotherapy at the time of the potential nodal recurrence. In our study, no lymph node metastases were found among 26 patients after the restaging surgery. This may be due to the structure of the study group. Patients with SCST (low risk of nodal involvement), and patients with dysgerminoma (high risk of nodal metastases) constituted 15 and 5 patients, respectively. Peritoneal staging was important because 3 out of 26 patients were upstaged; however, it influenced management only in 1 (3.85%) case. Among two patients, who were upstaged from IA to IC3, one patient with dysgerminoma received chemotherapy and one with granulosa cell tumor refused the adjuvant treatment. The patient diagnosed with dysgerminoma was upstaged from IC to IIB and would have received chemotherapy anyway.

Germ cell tumors are highly chemosensitive, which also supports their fertility-sparing management and accounts for a generally good prognosis. The increase in NEOC survival rates has been observed since the introduction of platinum-based chemotherapy [18]. The most commonly recommended regimen consists of bleomycin, etoposide, and cisplatin (BEP). The survival rates described in the literature reach 100% and 75% in the early and advanced stages, respectively [19]. The BEP regimen was also administered to 94% of the study group presented in the above research. Adjuvant chemotherapy regimens seem to have quite a low gonadotoxic effect, as the majority of women maintain their menstrual cycles in more than 85–90% of cases [8,20,21]. Nevertheless, the resumption of menstruation after oncologic treatment does not necessarily imply normal fertility. Highly gonadotoxic alkylating agents are not administered in cases of NEOC. Bleomycin and etoposide are related to a low (<20%) risk of gonadotoxicity according to the American Society of Clinical Oncology [22], while cisplatin has a higher risk of causing ovarian failure. Some clinicians replace cisplatin with carboplatin in order to diminish its toxicity in children. BEP can impair fertility due to follicle destruction and therefore the reduction of primordial follicles and ovarian stromal fibrosis. The effect of chemotherapy is directly related to the type of drug, schedule, total dose, and the duration of treatment [23,24,25]. In the presented study, 77% of the women who were qualified for chemotherapy were administered only three or four cycles. The moderate gonadotoxic effect of the chemotherapy in NEOC was confirmed in our study, as it was not related to childbearing. Similar observations were reported by Tangir et al., who did not find significant differences in the fertility potential between patients who received FSS + chemotherapy and FSS alone [26]. Zamani et al. demonstrated that pregnancy rates improved with decreasing the number of chemotherapy cycles [27]. However, in the detailed analysis (Table 4), the lowest birth rate (31.81%) was found in the women with SCST tumors, who underwent FSS and chemotherapy. This may suggest a negative impact of chemotherapy on the chance of childbearing, especially as these patients were not older than the other women in the study.

It is worth mentioning that apart from the applied regimen, the gonadotoxicity of chemotherapy depends on the age of patients and their initial ovarian reserve. Since the majority of NEOC are diagnosed in adolescents and young women, the influence of treatment is much less harmful than beyond the age of 40, where the reproductive potential is naturally decreased [22]. Genetic factors also may play a role. Germline pathogenic variants in BRCA genes have shown to be potentially associated with a reduced ovarian reserve at diagnosis [28].

Data about the reproductive and obstetric outcomes after FSS of NEOC are mostly retrospective and refer more to GCT than SCST. They are usually reassuring and support fertility-sparing management. Since NEOCs are rare malignant tumors, the reported conception rates are heterogenous and vary from 15 to 59%, while the pregnancy rates range from 67% to 100% [12,22]. In the presented study, 66 out of 146 women (birth rate: 45.2%) conceived and delivered during the follow-up period. As our data are of a retrospective nature, it is unknown if these women were treated using assisted reproductive technology (ART). Nevertheless, using their own gametes, ART can only be employed if their fertility is preserved. In the studied time period, oncofertility procedures were not that popular in Poland, especially in ovarian neoplasms; therefore, we can only extrapolate that not many women cryopreserved their oocytes (no reimbursement for such procedures in Poland). The majority of those pregnancies must have been natural conceptions. An unknown factor, however, could be the use of oocyte donation programs, but, again, no reimbursements for such programs exist in our country. Johansen et al. performed the first prospective study regarding both the safety and efficacy of FSS in stage I NEOC [6]. The authors included 73 women with complete data recruited between 2008 and 2015. During the follow-up time, 11 out of 57 (19.3%) patients conceived naturally and delivered at term, while 7 (12.3%) unsuccessfully attempted assisted reproduction (ART). Their children had no congenital malformations. Our data are of a retrospective nature; therefore, it is unknown if the women were treated using ART, and no information regarding congenital malformations was available. Almost 94% of all reported deliveries occurred at term, and the majority of the neonates were born with good general condition. Zanetta et al. reported a series of 169 patients diagnosed with NEOC between 1982 and 1996, of whom 138 (81%) underwent fertility-sparing treatment [29]. The authors described 55 pregnancies among 32 women. At the same time, 128 patients maintained their menstruation. Yang et al. published a series of patients with NEOC treated with FSS (148 women) followed by adjuvant chemotherapy in 129 cases [30]. All of the women maintained menstruation and 79.5% of those who desired pregnancy (25 out of 44) delivered infants without congenital anomalies. We believe that the retrospective evaluation of the desire to be pregnant may cause some bias [31] and, for this reason, we presented the pregnancy rate of the entire fertility sparing subset of patients, which in our opinion seems to be a more appropriate demonstration of the data.

Once remission is achieved, women may start planning their pregnancy. However, the optimal time for conception is unknown. The longer the time since the treatment, the lower the risk of recurrence, but female fertility also decreases with age. Therefore, it is crucial to choose the right moment for childbearing. Based on our results, we suggest that the appropriate time window between the diagnosis and subsequent pregnancy might be 2 years. Women should be advised to wait at least 2 years before becoming pregnant, because this period is associated with the highest risk of recurrence.

## 5. Conclusions

To summarize, according to the actual data it is acceptable to offer fertility-sparing treatment to patients with germ cell tumors even in advanced stages, taking into account their high sensitivity to chemotherapy, the young age of the patients and whether they have a good prognosis. Fertility-sparing options may be safely proposed to young patients with sex cord-stromal tumors in stage I. However, the evidence concerning the safety of these procedures is still not well investigated, because of the rarity of such tumors and the fact that the data are exclusively retrospective with a limited number of patients. Although restaging surgery may change the patients’ stage, it has limited impact on the patients’ prognosis and survival. Therefore, it should be considered only in patients for whom it will significantly affect further treatment. Patients with NEOC aged younger than 22.6 are most likely to deliver a child after the treatment. Although patients aged ≥ 22.6 are still considered to be young women, they should be referred for reproductive consultations. The strength of our study is that a large number of patients were treated in one oncological center; thus, an expert pathologist could perform a second histopathologic diagnosis in difficult cases and we could ensure a long follow-up.

## Figures and Tables

**Figure 1 cancers-15-04170-f001:**
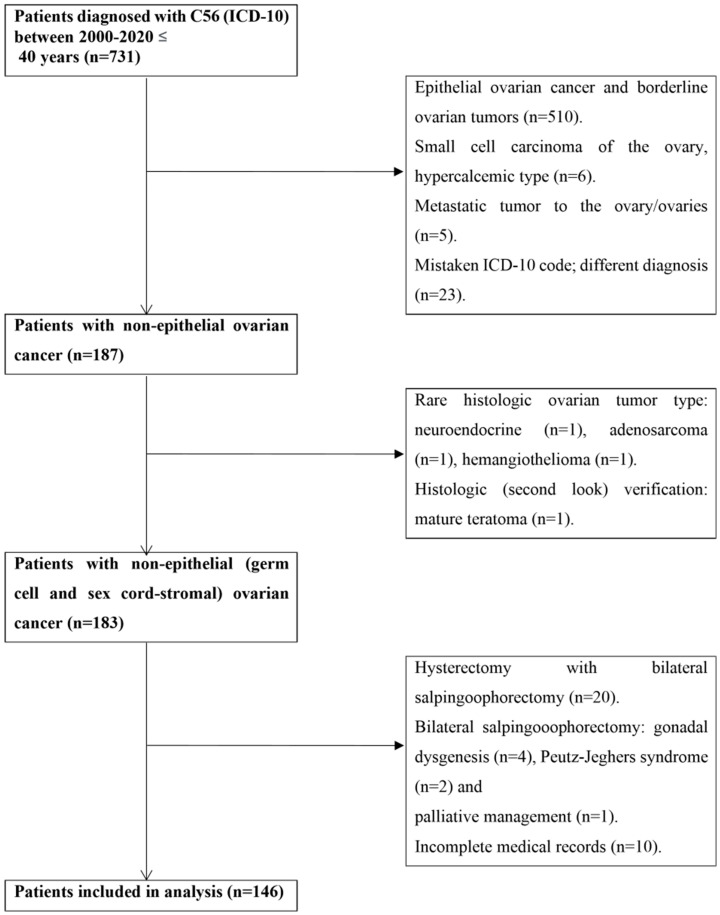
The flowchart of the study.

**Figure 2 cancers-15-04170-f002:**
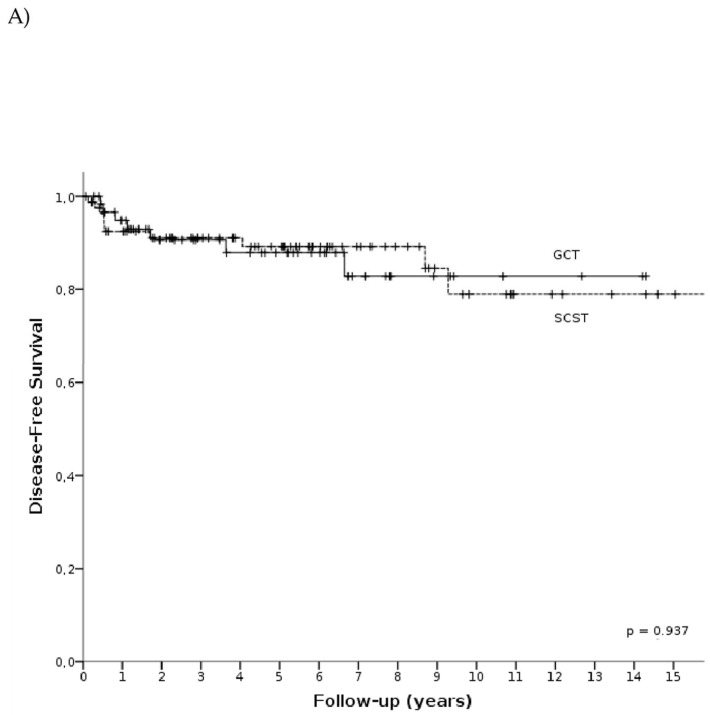
Disease-free survival (DFS) in patients according to histopathological examination (**A**), staging (**B**) and restaging surgery (**C**).

**Figure 3 cancers-15-04170-f003:**
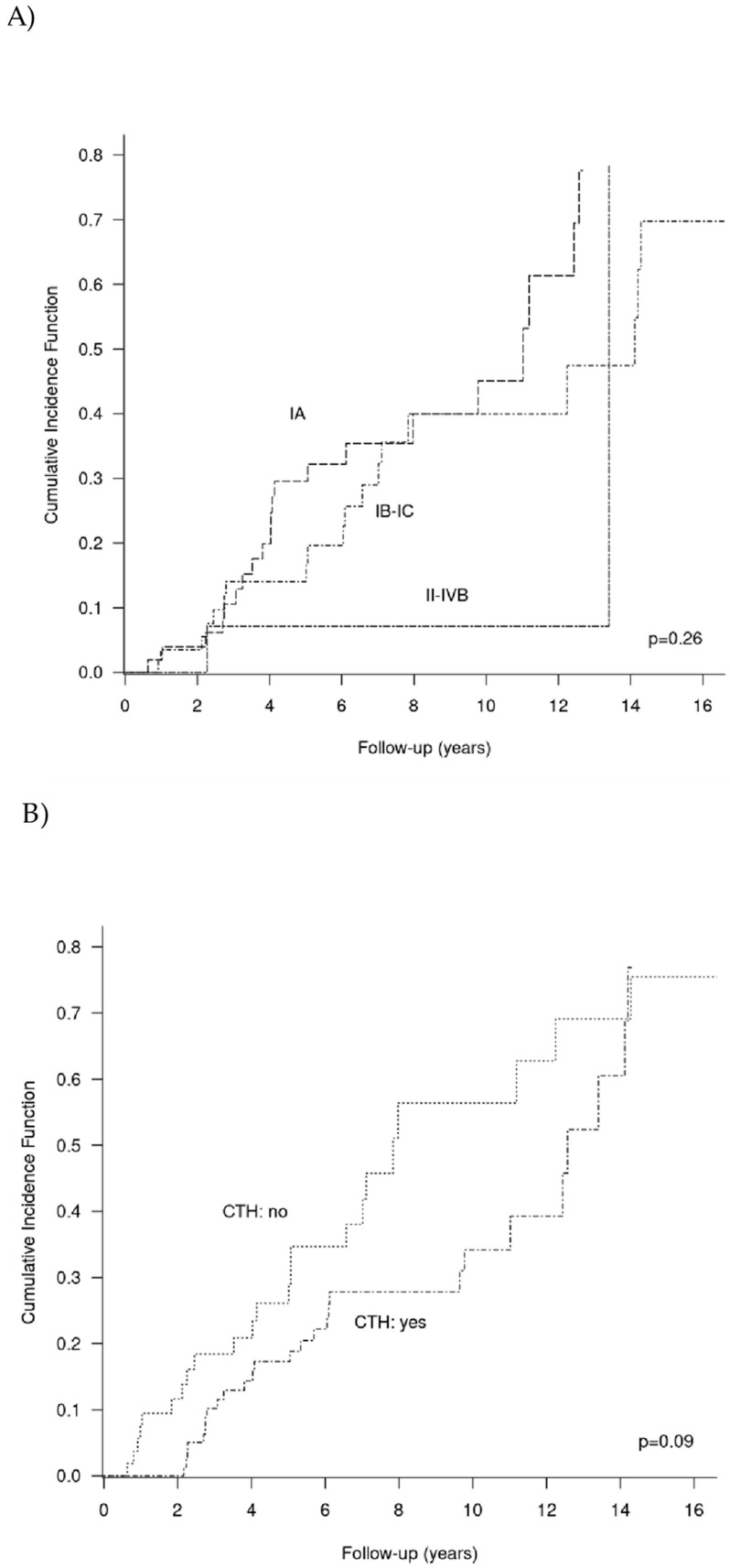
Cumulative incidence rates of childbearing according to: (**A**) stage, (**B**) chemotherapy (CTH: chemotherapy), and (**C**) age.

**Figure 4 cancers-15-04170-f004:**
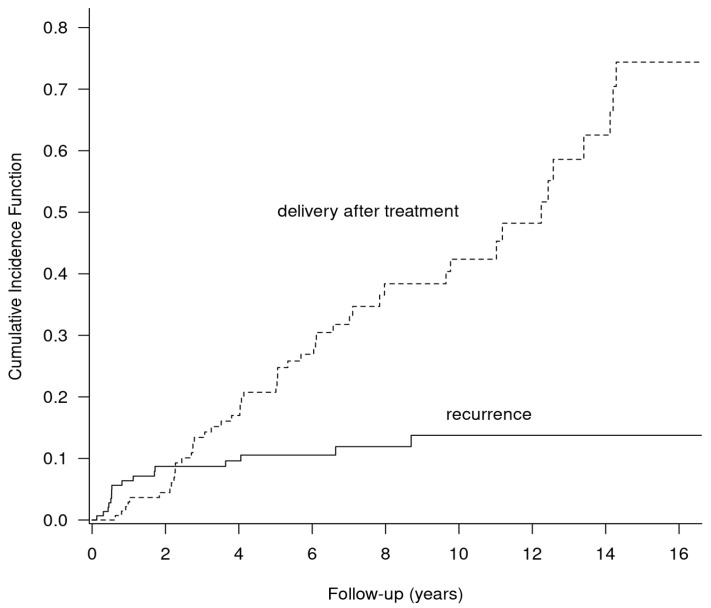
Cumulative incidence rates of childbearing and recurrence in patients with NEOC after FSM.

**Table 1 cancers-15-04170-t001:** Detailed characteristics of the study group (n = 146). FIGO stage included three up-staged patients after restaging surgery.

Variable	n = 146 (100%)	GCT (n = 84; 100%)	SCST (n = 62, 100%)
Histology		Dysgerminoma (n = 26, 30.95%)	Granulosa cell tumor (n = 46, 74.19%)
		Immature teratoma (n = 25, 29.76%)	Sertoli-Leydig cell tumor (n = 13, 20.97%)
		Mixed GCT (n = 19, 22.62%)	Mixed SCST (n = 1, 1.61%)
		Yolk sac tumor (n = 11, 13.1%)	Other (n = 2, 3.23%)
		Other (n = 3, 3.57%)	
Age at diagnosis (years)			
<25	64 (43.84%)	46 (54.76%)	18 (29.03%)
25–30	44 (30.14%)	29 (34.53%)	15 (24.19%)
31–35	26 (17.81%)	5 (5.95%)	21 (33.87%)
36–40	12 (8.21%)	4 (4.76%)	8 (12.91%)
FIGO stage			
I	133 (91.09%)	74 (88.09%)	59 (95.16%)
IA	65 (44.52%)	38 (45.24%)	27 (43.55%)
IB	1 (0.68%)	1 (1.19%)	0
IC IC1 IC2 IC3 IC (not specified)	51 (34.93%) 20 (13.69%) 10 (6.85%) 5 (3.42%) 16 (10.96%)	28 (33.33%) 5 (5.95%) 7 (8.33%) 3 (3.57%) 13 (15.48%)	23 (37.09%) 15 (24.19%) 3 (4.84%) 2 (3.22%) 3 (4.84%)
I (not specified)	16 (10.96%)	5 (5.95%)	9 (14.51%)
II IIA IIB	3 (2.05%) 1 (0.68%) 2 (1.37%)	3 (3.57%) 1 (1.19%) 2 (2.38%)	0
III IIIA IIIB IIIC III (not specified)	9 (6.16%) 3 (2.05%) 3 (2.05%) 2 (1.37%) 1 (0.68%)	7 (8.33%) 3 (3.57%) 2 (2.38%) 1 (1.19%) 1 (1.19%)	2 (3.22%) 0 1 (1.61%) 1 (1.61%) 0
IVB	1 (0.68%)	1 (1.19%)	0
Primary surgical approach			
Laparotomy	96 (65.75%)	71 (84.52%)	25 (40.32%)
Laparoscopy	39 (26.71%)	12 (14.29%)	27 (43.55%)
Unknown	11 (7.53%)	1 (1.19%)	10 (16.13%)
Type of primary surgery			
Cystectomy/tumorectomy	15 (10.27%)	13 (15.48%)	2 (3.22%)
Adnexectomy	120 (82.2%)	63 (75%)	57 (91.94%)
Adnexectomy with contralateral cystectomy	11 (7.53%)	8 (9.52%)	3 (4.84%)
Restaging surgery	26 (17.81%)	13 (15.48%)	13 (20.97%)
Pelvic lymphadenectomy *	33 (22.6%)	22 (26.19%)	11 (17.74%)

* Applies to patients after primary and restaging surgery.

**Table 2 cancers-15-04170-t002:** Characteristics of patients after restaging surgery.

	Mean/Number (Range/%)
Age	27.62 (17–38)
Days between first and restaging surgery	63 (34–83)
FIGO stage primary/after restaging	
IA => IA	12 (46.15%)
IA => IC	2 (7.69%)
IC => IC	9 (34.62%)
IC => IIB	1 (3.85%)
I unspecified => I unspecified	2 (7.69%)
Histology	
Granulosa cell tumor	9 (34.62%)
Sertoli Leydig cell tumor	4 (15.38%)
Mixed SCST	2 (7.69%)
Dysgerminoma	5 (19.23%)
Immature teratoma	3 (11.54%)
Mixed GCT	3 (11.54%)

**Table 3 cancers-15-04170-t003:** Adjuvant chemotherapy in 86 patients with NEOC after FSM.

Stage Histology		Number of Patients with Adjuvant Chemotherapy (Yes/No)					
		IA	IB	IC	I (Not Specified)	≥II	Total
Germ cell tumor	Dysgerminoma	2/8	1/-	8/1	-/-	6/-	17/9
	Immature teratoma	9/5	-/-	6/-	2/-	3/-	20/5
	Mixed GCT	4/2		8/1	3/-	1/-	16/3
	Yolk sac tumor	6/1	-/-	3/-	-/-	1/-	10/1
	Other GCT	-/2	-/-	1/-	-/-	-/-	1/2
	Total	21/18	1/-	26/2	5/-	11/-	64/20
Sex cord-stromal tumor	Granulosa cell tumor	-/20	-/-	9/10	3/2	2/-	14/32
	Sertoli-Leydig	3/3	-/-	-/3	3/1	-/-	6/7
	Mixed SCST	1/-	-/-	-/-	-/-	-/-	1/-
	Other SCST	-/1	-/-	1/-	-/-	-/-	1/1
	Total	4/24	-/-	10/13	6/3	2/-	22/40

**Table 4 cancers-15-04170-t004:** Relationship between diagnosis, treatment, and obstetric results (CTH: chemotherapy).

Histology	Type of Treatment	Number of Patients	Number of Patients Who Gave at Least One Birth after Treatment (Birth Rate)	Number of Patients Who Gave at Least Two Births after Treatment	Number of Successful Pregnancies (Ending in Childbirth)	Mean Age (Years)
Germ cell tumor	FSS	20	10 (50%)	4 (20%)	14	26.8
	FSS + CTH	64	26 (40.63%)	12 (18.75%)	40	23.5
Sex cord- stromal tumor	FSS	40	23 (57.5%)	7 (17.5%)	32	29.2
	FSS + CTH	22	7 (31.81%)	2 (9.1%)	9	26

**Table 5 cancers-15-04170-t005:** Characteristics of newborns of patients with NEOC. Newborns of women diagnosed with NEOC during pregnancy are presented separately.

Variable	Children Born after Treatment (n = 96) *	Children Conceived before Diagnosis (n = 10)	
		Diagnosis during pregnancy	Diagnosis and delivery at the same time
Sex	Male = 54 Female = 42	Male = 1 Female = 2	Male = 2 Female = 5
Weight: Mean Standard deviation Range	3469 g 498.5 g 2160–4950 g	3520 g 533.3 g 2620–3640 g	2787 g 795.9 g 1600–3620 g
Delivery			
Preterm	6 (6.25%)	1 (33.33%)	4 (57.14%)
32 Hbd 34 Hbd 35 Hbd 36 Hbd	1 1 1 3	- - 1 -	3 - - 1
Full-term	90 (93.75%)	2 (66.67%)	3 (42.86%)
Apgar score (points)			
8–10	90 (94.75%)	2 (66.67%)	7 (100%)
4–7	1 (0.1%)	-	-
0–3	-	-	-
Unknown	5 (5.2%)	1 (33.33%)	-

* The number of successful pregnancies was 95 because one of them was a twin pregnancy.

## Data Availability

Data are available from the authors of the study upon request.

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
