# Peer review of "Obstetric Results after Fertility-Sparing Management of Non-Epithelial Ovarian Cancer"

_cancers, 2023, doi:10.3390/cancers15164170_

Round 1
Reviewer 1 Report (Previous Reviewer 1)
Nil
Reviewer 2 Report (Previous Reviewer 2)
The authors have considered the recommendations on first review and revised the manuscript accordingly, and it is now much improved in terms of language, clarity of presentation of results and overall layout.
The English language is much improved from the original submission. Only the occasional spelling / grammar error is identified.
This manuscript is a resubmission of an earlier submission. The following is a list of the peer review reports and author responses from that submission.
Round 1
Reviewer 1 Report
1. What is the appropriate time window between treatment course and first pregnancy? this may be also mentioned.
2. Is there any pregnancy after the treatment of recurrence of the NEOC?
3. The pelvic lymphadenectomy rate in the study group was only 22.6%,in consideration of comprehensive staging procedure, what is your opinion in the necessity of pelvic lymphadenectomy of the surgery of NEOC?
4. During the post-operative and post-chemotherapy follow up, other than the use of transvaginal ultrasonography, what kind of image study is performed in your study group?
5. To avoid the gonado-toxicity of chemotherapy and expect the future pregnancy, do you consider the temporary use of leuprorelin acetate during the chemotherapy for protection of ovarian function?
6. Is there any difference of the survival and obstetric outcome between the laparoscopic and laparotomic group?
Reviewer 2 Report
The authors have included non-epithelial ovarian tumours in the study, focusing on malignant tumours, but there are benign lesions included with FIGO staging applied. The classification of tumours and the terminology used needs revision and the intended to be included in the study revised accordingly.
The language needs revision. Even the names of tumours used need to be revised and the WHO classification terminology used consistently.
Reviewer 3 Report
This study was conducted to evaluate the obstetrical outcome in cases of non-epithelial ovarian cancer (NEOC) treated with fertility sparing management (FSM).
The authors retrospectively analyzed146 patients who were diagnosed NEOC, and cumulative incidence rate of childbearing was rising, without any relation of chemotherapy.The result of this study suggested avoiding pregnancy two years after treatment due to the risk of recurrence.
It seems to be valuable information, however, this kind of research was already reported by previous article (G Johansen et.al., Gynecol Oncol 2019). Furthermore, there is no information of assisted reproductive treatment or oocyte cryopreservation in this study, hence these technique can be administered in young patients with NEOC, for gonadotoxicity, and which may delay the start of chemotherapy, resulting in poor prognosis. Because of lack of novelty and poor study design, this article seems to be unsuitable for publication as it is. Further research with more reproductive and obstetrical information would provide valuable results to evaluate the efficacy of FST in cases of NEOC.
Major comments
1. The authors stated 146 patients were analyzed in this study, however, in Figure.1, the final number of patients included must be 156. The authors must confirm the accurate number of patients.
2. Table.1 shows the characteristics of the study groups. the authors stated some patients were diagnosed different stage (Page 6, 146-149). In Table1. the stage remained in the first state, or up-graded after restaging surgery?
3. The Tables were not clear. For example, primal surgery and restaging surgery should be divided clearly. Moreover, in Table4, child weight was shown in average or median value?
4. In Figure 2A, the comparison on disease free interval between sec-cord and germ cell tumor is not important, because the clinical significance in comparing outcome between two groups of different malignancy with different treatment is little.
5. In Table3, what does "CTH" mean?
6. In Figure3 (A), p value was lacking.
7. Figure 4 seem to be unnecessary, because the cumulative incidence rate of childbearing and recurrence are shown in Table 5.
English in this article is partially poor, and some sentence can not be understood for not correct grammars. The article should be checked by a native English speaker.